# Governance for injury care systems in Ghana, South Africa and Rwanda: development and pilot testing of an assessment tool

Maria Lisa Odland [ID],[1,2] Abdul-Malik Abdul-Latif,[3] Agnieszka Ignatowicz,[1] Abebe Bekele,[4] Kathryn Chu [ID],[5,6] Anthony Howard [ID],[7,8] Stephen Tabiri,[9,10] Jean Claude Byiringiro [ID],[11,12] Justine Davies,[1,13] On behalf of The Equi-Trauma Collaborative

For numbered affiliations see end of article.

**Correspondence to**
Dr Agnieszka Ignatowicz;
a.m.ignatowicz@bham.ac.uk

## ABSTRACT

**Objectives** This study aims to evaluate health systems governance for injury care in three sub-Saharan countries from policymakers' and injury care providers' perspectives.

**Setting** Ghana, Rwanda and South Africa.

**Design** Based on Siddiqi _et al_'s framework for governance, we developed an online assessment tool for health system governance for injury with 37 questions covering health policy and implementation under 10 overarching principles of strategic vision, participation and consensus orientation, rule of law, transparency, responsiveness of institutions, equity, effectiveness or efficiency, accountability, ethics and intelligence and information. A literature review was also done to support the scoring. We derived scores using two methods— investigator scores and respondent scores.

**Participants** The tool was sent out to purposively selected stakeholders, including policymakers and injury care providers in Ghana, Rwanda and South Africa. Data were collected between October 2020 and February 2021.

**Primary and secondary outcomes** Investigator-weighted and respondent percentage scores for health system governance for injury care. This was calculated for each country in total and per principle.

**Results** Rwanda had the highest overall investigator-weighted percentage score (70%), followed by South Africa (59%). Ghana had the lowest overall investigator score (48%). The overall results were similar for the respondent scores. Some areas, such as participation and consensus, scored high in all three countries, while other areas, such as transparency, scored very low.

**Conclusion** In this multicountry governance survey, we provide insight into and evaluation of health system governance for trauma in three low- and middle-income countries (LMICs) in sub-Saharan Africa. It highlights areas of improvement that need to be prioritised, such as transparency, to meet the high burden of trauma and injuries in LMICs.

## INTRODUCTION

Injury is a leading cause of disability globally and responsible for more than 5 million deaths each year.[1] Mortality from injury

## STRENGTHS AND LIMITATIONS OF THIS STUDY

⇒ This is the first study to use an adapted tool to assess health systems governance for injury care in low- and middle-income countries (LMICs).

⇒ We obtained responses from a range of professionals working with trauma care in three different countries in sub-Saharan Africa.

⇒ A major limitation is that we only had five participants in two of the countries (Rwanda and South Africa) and the low number of respondents could have introduced selection bias.

⇒ If there was no available evidence, the investigators had to weigh the replies from the respondents according to their background, which involved making assumptions about the respondents' knowledge of the subject.

⇒ This survey tool provides useful insight in the governance of trauma systems in three LMICs with different development status.

account for more deaths than tuberculosis (TB), malaria and HIV combined,[1] and 90% of these deaths occur in low- and middle-income countries (LMICs).[2] While deaths and disability-adjusted life years (DALYs) lost from many other conditions are in decline, DALYs from injuries remain stubbornly high. Indeed, deaths from injury are predicted to become the leading cause of death by 2030.[3] Despite improvements in road traffic safety in most high-income countries, many LMICs are now having an increasing number of motorised vehicles and road traffic accidents in addition to other common accidents causing injuries such as falls and burns. Still, only a few LMICs have well-defined trauma systems or trauma registries.[4]

The United Nations Development Programme defines governance as the exercise of political, economic and administrative

authority in managing a country's affairs at all levels.[5] Governance has long been a critical factor that influences a country's economic growth, social advancement and general development. It is recognised as especially important for advancing progress towards attaining the Sustainable Development Goals in LMICs.[6] Over recent years, there has been increasing interest in health systems governance with the recognition that good governance leads to better health outcomes for individuals and populations.[6] In 2014, the Lancet-University of Oslo Commission on Global Governance for Health called for a cross-sectoral global action and platform for health governance. This platform may serve as a policy forum to allow the contribution of diverse stakeholders to frame issues, set agendas and debate policies that affect health and health equity.[7] The WHO first introduced the term 'stewardship'—a part of governance, in the year 2000, and called for strategic policy frameworks that would allow the incorporation of effective oversight, regulation, incentives and accountability in health governance.[8]

Health system governance thus involves setting evidence-based shared strategic visions and objectives, in addition to making policies, legislation and deploying resources to ensure the goals and objectives are achieved.[9] However, despite its importance in supporting the delivery of better services and improved health outcomes, little is done to monitor and evaluate health system governance in LMICs.[10–12] Additionally, literature on health system governance around trauma care in LMICs is scarce.[13] Previous studies on health system governance have primarily focused on general health systems functions and particularly on the role of government in governance and the involvement of communities.[14] Moreover, the sparse disease-specific literature that exists focuses on the governance of programmes for immunisation,[15] TB control,[16] mental healthcare[17–19] and achieving global HIV goals.[20]

Given the prevalence of injuries and the recognised need to invest in health services to provide trauma care, good governance will be essential to ensure that the care provided is of high quality and accessible to those who need it. As part of a larger project that identified barriers in access to quality care for people who have been injured in LMICs,[21] we adapted a tool to assess the health system governance for trauma care in three diverse countries in sub-Saharan Africa—Ghana, Rwanda and South Africa. Our aim was to try to understand the foundations on which to build improved health systems for trauma and injuries in LMICs.

## METHODS
### Study setting
The study was conducted in three LMICs in sub-Saharan Africa: Ghana, Rwanda and South Africa which have vastly different development and health systems. Ghana is a lower-middle-income country, with an estimated population of 30.4 million people (2019), a life expectancy of 63.8 years and a gross national income (GNI) per capita of $2220.[22] While health service delivery in the country is largely provided by government, private health institutions also provide significant proportion of health services to the population.[23] The National Ambulance Service provides 24-hour prehospital care for accidents and emergencies as part of the care provided by the government.[23] It has been estimated that 7.56% of deaths and 7.24% of DALYs in Ghana are due to trauma.[22 24]

Rwanda, a landlocked East African country of 12.6 million people, has a life expectancy of 68.7 years and GNI per capita of $830.[22] It is classified as a low-income country. Around 9% of all deaths and 10% of DALYs are due to trauma.[22] Following the near decimation of its health system by the 1994 genocide, the country has taken steps to strengthen it, giving autonomy to District Health Services to serve urban and rural zones.[25] It introduced the Community-Based Health Insurance system in 1999/2000 to provide health insurance to rural populations.[26] However, the health system is still challenged, and deficiencies exist in the provision of quality trauma care.[27]

South Africa is an upper-middle-income country, with a population of 68.6 million and a life expectancy of 63.8 years. Injuries are estimated to be responsible for 10% of death and 11% of DALYs.[22 24] South Africa has the third biggest economy in Africa and a GNI per capita of $6040.[22] Most South Africans (84%) access health services through government clinics, while the more affluent people go to private hospitals.[28]

### Data collection
Building on the framework and tool developed by Siddiqi et al[6] for assessing the health system governance in developing countries, we developed an assessment tool for injury/trauma health system governance with 37 questions covering health policy and implementation using the 10 overarching principles outlined by Siddiqi et al: strategic vision, participation and consensus orientation, rule of law, transparency, responsiveness of institutions, equity, effectiveness or efficiency, accountability, ethics and intelligence and information (online supplemental appendix 1 and table 1). Adjustments to the original tool were made to tailor the questions to trauma; these were made based on discussions between the authors of this paper. The resultant tool was piloted for acceptability and comprehensibility before use. Data were collected over a 5-month period from October 2020 to February 2021 by participants self-completing an online Word or Google form, based on their preference. Participants were requested to select a single response for each question and included a free-text field for notes, and provision of evidence to support their responses was encouraged. All responses were imported and analysed in Microsoft Excel.

An extensive review of grey and published literature documents was also done to support the assessment, particularly the scoring.

**Table 1** Applying Siddiqi *et al*'s governance framework to trauma care systems in Ghana, Rwanda and South Africa

| S/N | Governance principle | Explanation of principles based on Siddiqi *et al*'s framework | Domain captured for trauma care | Maximum score for principle |
|---|---|---|---|---|
| 1 | Strategic vision | Through an understanding of the historical, cultural and social complexities of society, leaders have a strong sense of direction for the achievement of long and broad health and human development goals. | There is a detailed long-term strategic plans to improve trauma care. | 12 |
| 2 | Participation and consensus orientation | Everyone or interest groups or institutions acting on behalf of everyone should be given the chance to have a say in relation to decisions about health. This is built on the principle of freedom of association and speech as well as capacities to participate constructively. Good governance should be able to mediate between differing opinions among stakeholders on health, policies and procedures in order to reach a mutual understanding that is beneficial for all. | There is stakeholder participation and level of engagement in policy formulation and implementation for trauma. | 3 |
| 3 | Rule of law | Legal frameworks or policies relating to human rights on health especially should be applied impartially. | There is availability and enforcement of laws, guidelines and policies to support trauma care. | 6 |
| 4 | Transparency | There should be free flow of information on all health matters. There should be enough information available to all to not only monitor but also understand health matters. Processes, institutions and information should be directly accessible to those concerned with them. | There is transparency on commitments to trauma and available information on indicators and other trauma related information for providers (district) involved in local trauma service provision. | 3 |
| 5 | Responsiveness | Institutions and processes should promptly serve all stakeholders and ensure that their health and non-health needs are met without delays. | Trauma systems are responsive to trauma care needs of the population. | 10 |
| 6 | Equity and inclusiveness | Everyone should have the opportunity to improve or maintain their health and well-being. | There is equity in access to quality trauma care. | 8 |
| 7 | Effectiveness and efficiency | Institutions and processes should maximise available resources to render best healthcare services according to population needs, as well as influence improved health outcomes. | There is the existence of organisational capacity including human resource, communication processes to support quality trauma provision. | 6 |
| 8 | Accountability | People put in positions of trust from government, the private sector and civil society organisations should be accountable to the public and institutional stakeholders. Accountability in this sense varies depending on the type of institution or organisation and whether or not decisions are for internal or external purposes. | There is evidence of accountability between service providers and users in the provision of trauma care. | 3 |
| 9 | Intelligence and information | Essentials for understanding of the health system to guide the implementation of good policies that are based on empirical data to influence the behaviour of different interest groups that support the strategic vison for health. | There is availability of tools and capacity to capture trauma care data. | 2 |
| 10 | Ethics | Widely accepted principles of healthcare ethics: non-maleficence, beneficence and justice. This also includes ethics in healthcare research essential to safeguard the interest and rights of the patients. | There is enforcement of high ethical standards in trauma care provision and research. | 3 |
| | **Maximum score** | | | **56** |

Questions for each governance principle are in table 3.

## Survey respondent selection

Our aim was to recruit participants from health policy or senior leaders in trauma care provision in each country. Given that we expected potential participants to have sound knowledge of the policy and governance context for injury care in their countries, we aimed for a sample size of 5–8 respondents. We contacted potential participants until at least the minimum number was achieved. A combination of purposive and snowball sampling was used to recruit respondents, with potential participants identified with the support of in-country senior researchers within injury care. Emails were sent to potential participants to request their participation in the study; for each invited participant, two further reminders were sent.

## Grey literature search

We also searched for and reviewed programme documents, policies, annual reports and standard operating procedures for each country. Searching was done through the websites of government organisations at the national and subnational level, websites of international organisations and the Google search engine. The search terms included the country name and trauma policy, trauma

law, strategic plan for trauma, injury, injury work plan, injury policy, injury care, trauma care, injury guidelines, trauma guidelines and combinations of these. There was no restriction on the year of publication.

## Scoring

Scoring was done separately for each country. For each principle, there were already a set number of questions outlined by Siddiqi *et al* to give a maximum score. Responses were awarded points for each question and treated as binary categorical (0 or 1) or ordinal (0, 1 or 2) (see online supplemental appendices). We derived scores using two methods—investigator-weighted scores and respondent scores. For the respondent scores, the mean score across respondents for each question was computed as the average score from the responses for each question. Given that the response rate for each country differed, the denominator (n) varied based on the number of responses: 11 for Ghana, 5 for Rwanda and 5 for South Africa.

While the investigator-weighted scores considered the following to derive a final score for each question: results from respondents, respondents' professional roles and the availability of evidence from policy documents and the grey literature searches. These investigator scores were derived after discussions between the authors. Consideration of the respondent's professional roles depended on the question asked; more emphasis was given to responses from policymakers rather than trauma care providers for policy-related questions, and more was given to trauma care providers for questions related to service provision. So, for example, if a trauma care provider gave a score of 0, and the policymaker gave a score of 2 on a question related to policy, such as 'are there legal documents of injury care?', the question would receive a final score of 2 as the policymaker was more likely to have up-to-date knowledge. If a policy document was available to answer a question definitively, the literature took precedence over respondents. This process was done through discussions between two authors: A-MA-L and MLO. When there were disagreements, third and fourth investigators served as arbiters (AI and JD).

Both investigator scores and the average respondent scores for each principle were calculated by dividing the achieved score in each country by the total score possible to achieve and multiplying it by 100. Comparisons across countries are described for each of the 10 principles and overall.

## Patient and public involvement

It was not appropriate or possible to involve patients or the public in the design, or conduct, or reporting, or dissemination plans of our research.

## RESULTS

Table 2 shows a breakdown of the number of respondents from each country and their employment role at the time of completing the survey. Respondents were made up of key officials employed directly by or advising Department or Ministry of Health, trauma care providers (some of whom were also involved in research) and government officials. Thirteen potential respondents were contacted from each country.

Online supplemental appendix table 2 shows the investigator score for each country according to each question and percentage score for each principle and overall for each country. Some of the respondents provided evidence to support their answers such as policy documents and peer-reviewed papers.

Rwanda had the highest overall investigator percentage score (70%) followed by South Africa (59%). Ghana had the lowest overall investigator percentage score (48%) (table 3). The overall results were similar for the respondent average percentage score, with Rwanda scoring 39.85 (71%) in total, South Africa 31.07 (56%) and Ghana 18.5 (33%) (online supplemental appendix table 3—with both percentage scores shown for comparison).

Considering the investigator scores, Rwanda had the highest scores for each principle except for equity. Participation and consensus, in particular had a very high score in Rwanda (100%), while the other scores were between 70% and 80%, apart from strategic vision (66.7%) and equity (37.5%). Like Rwanda, South Africa also had high investigator-weighted scores overall but had low scores for strategic vision (50.0%), equity (37.5%) and intelligence and information (50.0%). For transparency, South Africa had a score of 0%. On the other hand, South Africa had high scores for participation and consensus orientation (100%), rule of law (83.3%), accountability (100%) and ethics (100%). Responsiveness of institutions (50.0%) and effectiveness and efficiency (66.7% and 66.7%) received medium-high scores. Ghana's highest scores were for the principles of rule of law (83.30%), effectiveness and efficiency (66.70%) and ethics (66.70%). However, the scores were low for the other principles, especially strategic vision (33.30%), transparency (0%),

| S/N | Country | Potential participants contacted (n) | Respondents (n) | Policy respondents (n) | Trauma care providers (n) |
|-----|---------|--------------------------------------|-----------------|------------------------|---------------------------|
| 1 | Ghana | 13 | 11 | 3 | 8 |
| 2 | Rwanda | 13 | 5 | 3 | 2 |
| 3 | South Africa | 13 | 5 | 3 | 2 |

**Table 2** Breakdown of respondents from each country and their characteristics

**Table 3** Investigator score for each question and percentage score for each principle and overall for Rwanda, Ghana and South Africa, respectively

| Principle | One question out of many questions asked in this principles | Maximum score for questions | Rwanda score | Ghana score | South Africa score |
|---|---|---|---|---|---|
| Strategic vision | Is there specific mention of trauma in the national health plan or policy? Or are there specific national health policies around trauma care? | 12 | 8 (67%) | 4 (33%) | 6 (50%) |
| Participation and consensus | What is the level of stakeholder engagement/community inparticipation at the national and provincial level in trauma policy and related interventions? | 3 | 3 (100%) | 3 (100%) | 3 (100%) |
| Rule of law | Are there guidelines for accreditation of trauma care providers (doctors, nurses, etc) and are these enforced? | 6 | 4 (67%) | 5 (83%) | 5 (83%) |
| Transparency | Are managers (district directors of health, medical superintendents of hospitals) evaluated on their health facility or facilities reaching specific targets for trauma care? And if so, are the results of these evaluations available and accessible? | 3 | 3 (100%) | 0 (0%) | 0 (0%) |
| Responsiveness of institutions | Is there mandatory reporting of health facility trauma data and is this used to define the burden of injury at a national level? | 10 | 8 (80%) | 5 (50%) | 5 (50%) |
| Equity | Are there national level financial schemes to ensure the poor who are injured do not have to pay out of pocket direct medical costs of trauma care? | 8 | 3 (38%) | 2 (25%) | 3 (38%) |
| Effectiveness and efficacy | Is there a national trauma registry (information management for trauma care)? Is it used? In both private and public? | 6 | 5 (83%) | 4 (67%) | 4 (67%) |
| Accountability | Are there mechanisms to report failing trauma services to policy makers or regulatory authorities? | 3 | 1 (33.3%) | 1 (33.3%) | 3 (100%) |
| Intelligence and information | Do staff providing trauma services understand what data need to be captured and do they have the right data capturing tools to enable them to do this? | 3 | 1 (33%) | 1 (33%) | 1 (33%) |
| Ethics | Are there any standard operating procedures in place to ensure quality and ethical trauma care for injured people? | 3 | 3 (100%) | 2 (67%) | 3 (100%) |
| **Overall total (% maximum overall score)** | | **56** | **39 (70%)** | **27 (48%)** | **33 (59%)** |

equity (25%) and accountability (33.30%). This gave Ghana the overall lowest score in the governance assessment for trauma with an investigator score of 48.20% (see table 3). The only principle that received a 100% investigator score in all the countries was participation and consensus orientation.

Discrepancies between investigator scores and average respondent scores were mostly seen in Ghana, where the overall scores were 33.0% versus 48.20%, respectively (table 4 and online supplemental appendix tables). There were fairly large discrepancies for almost all the principles except for equity (17.5% vs 25.0%), effectiveness and efficiency (54.7% vs 66.70%) and accountability (28.70% vs 33.30%) (online supplemental appendix 2). The average respondent and investigator percentage scores for each principle were more similar for the other two countries. In Rwanda, the overall average respondent percentage score was 71.2%, and the average percentage investigator score was 69.6%. Most of the individual principles had similar respondent percentage scores except for transparency (60.0% vs 100.0%), accountability (70.0% vs 33.3%) and intelligence and information (80.0% vs 50.0%). In South Africa, the overall average respondent percentage score was 55.5%, and the overall investigator percentage

score was 58.9%. Similar to Rwanda, the individual principle scores were more or less similar except for those of strategic vision (38.9% vs 50.0%), accountability (89.0% vs 100.0%), intelligence and information (37.5% and 50.0%) and ethics (89.0% vs 100%).

## DISCUSSION

To the best of our knowledge, this is the first study that has assessed governance for trauma health systems across multiple countries. The application of our adapted tool revealed strengths and weaknesses in policies and governance of trauma care in Ghana, South Africa and Rwanda. Rwanda achieved fairly high scores (70%), compared with South Africa (59%), and Ghana, which had the lowest score (40%). However, considering the massive burden of injuries and trauma in these countries, our results suggest that there is room for improvement even in the higher-performing countries. At the same time, the gap between the burden of disease and available governance systems and structures was especially seen in Ghana. The benefits in policies can be seen when considering the free maternal healthcare policy which has been vital in

Table 4 Summary results by principle for each country individually including achieved percentage score (average score and investigator score)

| Principle | Maximum scores | Rwanda | | Ghana | | South Africa | |
|---|---|---|---|---|---|---|---|
| | | % achieved (respondent scores) | % achieved (investigator scores) | % achieved (respondent scores) | % achieved (investigator scores) | % achieved (respondent scores) | % achieved (investigator scores) |
| Strategic vision | 12 | 67.1 | 66.7 | 17.30 | 33.30 | 38.9 | 50.0 |
| Participation and consensus orientation | 3 | 100.0 | 100.0 | 46.70 | 100 | 100.0 | 100.0 |
| Rule of law | 6 | 79.2 | 66.7 | 63.20 | 83.30 | 80.5 | 83.3 |
| Transparency | 3 | 60.0 | 100.0 | 16.30 | 0 | 0.0 | 0.0 |
| Responsiveness of institutions | 10 | 76.5 | 80.0 | 35.40 | 50 | 52.5 | 50.0 |
| Equity | 8 | 36.9 | 37.5 | 17.50 | 25 | 40.4 | 37.5 |
| Effectiveness and efficiency | 6 | 89.2 | 83.3 | 54.70 | 66.70 | 66.7 | 66.7 |
| Accountability | 3 | 70.0 | 33.3 | 28.70 | 33.30 | 89.0 | 100.0 |
| Intelligence and information | 2 | 80.0 | 50.0 | 36.50 | 50 | 37.5 | 50.0 |
| Ethics | 3 | 86.7 | 100.0 | 52.30 | 66.70 | 89.0 | 100.0 |
| **Overall score** | **56** | **71.2** | **69.6** | **33.00** | **48.20** | **55.5** | **58.9** |

ensuring access to healthcare for women and children, but policies do not exist for injuries and trauma care.[29]

Rwanda scored relatively highly in our survey. This could be because having successfully achieved the MDGs (Millennium Development Goals), Rwanda has committed to reducing morbidity and mortality due to injuries.[8 30] This includes developing policies, training healthcare providers, investing in data collection and hosting its first national symposium on trauma and injuries in 2019.[31] Hence, there has been a focus on improving health systems to care for patients with injuries in the last few years. There is still high mortality and morbidity from injuries in the country. Still, interventions following recent policies and prioritisation of trauma care coupled with efforts to prevent injuries, for example, the recent introduction of speed cameras in urban areas, will likely improve the situation in the coming years.

Given the level of development—being the only upper-middle-income country in our study, it is surprising that South Africa had mediocre percentage scores of around 50%. Many LMICs have a high burden of injuries and trauma, but South Africa has a relatively large burden of homicide, violence and stabbings.[32] In addition to this, there are other common injuries, such as road traffic accidents and burns. Even though there are programmes, services and ongoing research on this topic, government stewardship and leadership has been absent.[32] Prevention of violence and injury should be a strategic priority for government programmes and policies, and this requires governance and leadership; there are valuable lessons that South Africa can learn from its own excellent governance structures for HIV care.[33]

Overall our results emphasise that more efforts are needed to strengthen overall governance for injury care, considering how crucial governance is to achieve Universal Health Coverage.[34] Finance cannot be neglected in this process. However, it is also critical to focus on the principles that were particularly weak in this study (transparency, accountability and intelligence and information), to improve the effectiveness of the health sector.[34] In particular, accountability and the correction of trauma care underperformance will remain issues without adequate data generation. On the other hand, WHO has developed a trauma registry for LMIC settings that can be tailored to individual country needs, uptake at national levels is lacking, and the use of data collected for health service quality improvement is underdeveloped. Rwanda is the only country in our study that uses the WHO-based trauma registry, and this is only used in five hospitals and without an active quality improvement programme, although there are plans to develop this.[30]

Another thing that was evident in our findings was the difference between the investigator and respondent scores. Rwanda had the highest score regardless of the scoring system used, and the overall investigator and respondent scores were similar. However, in the other two countries, the respondent score was lower than the investigator score, especially in Ghana, which had the lowest scores altogether. The difference between the respondent and investigator scores suggests that many respondents are unaware of relevant policies/governance structures for trauma in their respective countries. Awareness of these is the first step to using them in order to improve injury and trauma care in the respective countries.[35] Policies are useless if the people in charge of implementing them are unaware of them. According to our survey, this is mostly an issue in Ghana, but also somewhat in South Africa.

This study also revealed some interesting findings in relation to 'participation and consensus orientation', as it was the only principle where all three countries scored 100%. More involvement of stakeholders may improve

service delivery and reduce barriers to accessing quality care for injuries after trauma. But, this is not necessarily the case, as seen in Nigeria, where an increased involvement of stakeholders in the formulation and implementation of TB policies did not necessarily result in good TB control in the community and the health services in the country.[16]

It is likely that multiple components of governance need to be in place—in combination with the awareness of these—for the improvement of healthcare systems. For example, in Ethiopia, improved health system governance was expected to impact critically on scaling up mental healthcare within primary care facilities.[17] The presence of high-level government support was thought to be a strength along with a National Mental Health Strategy. But unfortunately, there was still a very low baseline awareness of mental healthcare planning and a lack of leadership and coordination of mental health planning at the national and district level. Indeed, a qualitative study using Siddiqi *et al*'s framework for mental health governance in South Africa found that facilitating factors to implementing integrated mental healthcare were using task-sharing models and establishment of district mental health teams to facilitate the development, and implementation of mental healthcare plans. The challenges were weak managerial and planning capacity to develop healthcare at the provincial and district level. All of which speak to the need for knowledge and implementation of governance structures for the improvement of healthcare. Hence to strengthen healthcare delivery, there is a critical need to strengthen leadership and coordination, and implementation at all levels; national, regional, district and down to individual healthcare facilities. There are valuable lessons from these other disease areas that can be used for governance structures to improve trauma care systems.

In this survey, we managed to obtain responses from a range of professionals working with trauma care in three different countries in sub-Saharan Africa. However, a major limitation is that we only had five participants in two of the countries (Rwanda and South Africa) and found soliciting the involvement of respondents difficult, despite having researchers with links to policy makers leading the study in each country. The low number of respondents could have introduced selection bias. We tried to overcome this bias using an investigator score. However, we may have found different results if we had achieved greater numbers of participants from each country. Nevertheless, our results have face validity, considering that injury care has been an area of focus in Rwanda,[31] and Rwanda scored highest in our governance survey. The investigator scores also had their limitations. If there was no available evidence, the investigators had to weigh the replies from the respondents according to their background, which involved making assumptions about the respondents' knowledge of the subject. We did our best to make sure the investigator scores were correct by checking the grey literature and available information.

Our scoring system has not been validated and we cannot be certain that the scores were always reflective of the true trauma systems governance of that country, or that one country is doing better than the other. Another limitation was that there was only one question focusing on injury prevention in our survey.

Nevertheless, our study is novel in looking at governance assessment for injuries in LMICs. This survey tool provides useful insight in the governance of trauma systems in three LMICs with different development status and provides evidence that governance systems for trauma need to be improved in certain areas in order to face the high burden of injuries in LMICs in the years to come.

## CONCLUSIONS

In this multicountry governance survey, we have shown that the governance structures for trauma are limited in three different countries in sub-Saharan Africa; Ghana, Rwanda and South Africa. Some areas, such as participation and consensus, scored high in all three countries, while other areas, such as transparency, scored very low. This study provides insight into the governance of trauma systems in these three countries and highlights areas that need to be prioritised in the years to come in order to meet the high burden of trauma and injuries. Assessment of the health systems governance for trauma, as we did in this study, provides evidence that should not only stimulate more research in this area but also support advocacy efforts to advance trauma care systems.

**Author affiliations**
[1]University of Birmingham, Institute of Applied Health Research, Birmingham, UK
[2]Department of Obstetrics and Gynecology, St Olavs Hospital Trondheim University Hospital, Trondheim, Trøndelag, Norway
[3]Volta Regional Health Directorate, Ghana Health Service, Accra, Greater Accra, Ghana
[4]University of Global Health Equity, Kigali, Gasabo, Rwanda
[5]Centre for Global Surgery, Department of Global Health, Stellenbosch University Faculty of Medicine and Health Sciences, Cape Town, South Africa
[6]Department of Surgery, University of Botswana, Gaborone, Botswana
[7]University of Leeds, Leeds Institute of Rheumatic and Musculoskeletal Medicine, Leeds, UK
[8]Nuffield Department of Orthopaedics, Rheumatology and Musculoskeletal Sciences, National Institute of Health Research (NIHR) Biomedical Centre, University of Oxford, Oxford, UK
[9]Ghana Hub of NIHR Global Surgery, Tamale, Northern, Ghana
[10]Department of Surgery, Tamale Teaching Hospital, Tamale, Northern, Ghana
[11]University of Rwanda College of Medicine and Health Sciences, Kigali, Rwanda
[12]Department of Surgery, University Teaching Hospital of Kigali, Kigali, Rwanda
[13]Centre for Global Surgery, Department of Global Health, Stellenbosch University Faculty of Medicine and Health Sciences, Cape Town, Western Cape, South Africa

**Acknowledgements** We would like to acknowledge to all of the participants in our study, who generously shared their time, experiences and insights with us.

**Contributors** The study was designed by MLO, A-MA-L, AI and JD. MLO, A-MA-L, KC, AB, ST and JCB contributed to the data collection. MLO, A-MA-L, AI and JD did the analysis and interpretation of results. MLO, A-MA-L, AI and JD drafted the manuscript. All authors reviewed the results and approved the final version of the manuscript. JD is the guarantor.

**Funding** This research was funded by the National Institute for Health and Care Research grant number 130036.

**Competing interests**  None declared.

**Patient and public involvement**  Patients and/or the public were not involved in the design, or conduct, or reporting, or dissemination plans of this research.

**Patient consent for publication**  Not applicable.

**Ethics approval**  The overall study was approved by University of Birmingham Research Ethics Committee, UK (ERN_20-00880).

**Provenance and peer review**  Not commissioned; externally peer reviewed.

**Data availability statement**  Data are available upon reasonable request.

**ORCID iDs**
Maria Lisa Odland http://orcid.org/0000-0003-4340-7145
Kathryn Chu http://orcid.org/0000-0002-8923-7447
Anthony Howard http://orcid.org/0000-0001-7746-1268
Jean Claude Byiringiro http://orcid.org/0000-0002-6445-1797

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
