## [Reviewer comments · BMJ Open]

ARTICLE DETAILS

TITLE (PROVISIONAL)	Governance for injury care systems in Ghana, South Africa and Rwanda: development and pilot testing of an assessment tool
AUTHORS	The Equi-Trauma, Collaborative; Odland, Maria Lisa; Abdul-Latif, Abdul-Malik; Ignatowicz, Agnieszka; Bekele, Abebe; Chu, Kathryn; Howard, Anthony; Stephen, Tabiri; BYIRINGIRO, JEAN CLAUDE; Davies, Justine

VERSION 1 – REVIEW

REVIEWER	J. Worlton, Tamara Uniformed Services University of the Health Sciences
REVIEW RETURNED	18-May-2023

GENERAL COMMENTS	Excellent paper and one that can be followed up in time to see if interventions/improvement has been accomplished. Some minor comments/edits: In abstract introduction- the second sentence does not seem to fit with the paper. You are evaluating this so perhaps this should read as little is known or published about healthcare governance. This study aims to evaluate governance in three sub-Saharan countries from policy maker and injury care provider perspective... Abstract conclusion: All three sentences are kind of saying the same thing. I recommend linking it back to your intro- This study provides baseline insight and evaluation of governanceAreas of improvement include Methods: The descriptions of the countries/study setting does not belong in methods- it should be in the introduction or discussion. If you have space for another table- I would put the country characteristics in a table form (study setting) that way the reader can see more clearly the differences between the three. I think table 1 is not needed and appendix table 1 should be used instead. When I looked at the table 1 it was not clear why each principle had a different max value. Is there a reason why only the investigator score is shown in Table 3 and both scores are only shown in appendix table 3? The level of concordance between investigator and respondent scores is a big point that should be highlighted more. I believe a bar graph could highlight this better. page 21 line 11- is this supposed to be Siddiqi vice Saddiqi?
--

REVIEWER	Glass, Nina Rutgers New Jersey Medical School
-----------------	--

GENERAL COMMENTS

Thank you for inviting me to review this manuscript. It is an important account that begins to address the topic of governance in the provision of trauma care. Overall I think that the manuscript is considered and well-written. I did have a few questions and a few editorial comments for the authors that I will outline here.

Abstract:

1. It seems as if maybe you were trying to cut too many words; e.g., "as" was omitted twice from the results section (should be "such as").
2. I will come to this again in the introduction, but you have not referred to the "increasing" burden of trauma in the abstract, so not sure if you can say that in the conclusion of the abstract ("high" or "overwhelming" or something else would be fine).
3. Just editorial, but the third bullet about how the study affects practice has a typo in the header

Introduction

4. Speaking of an increasing burden of traumatic injury always raises red flags for me; is injury burden actually increasing? Is its percentage of the cause of death and disability increasing because care for other conditions is improving faster? Aren't we driving safer cars and overall getting safer or is the burden of this condition actually increasing? I'm not sure if your overview addresses this (and maybe you don't have to and this is simply my problem).
5. The sentence in paragraph one beginning, "The Lancet-University of Oslo Commission. . ." is way too long and therefore something about it doesn't make sense; it may be a problem I saw several times in the manuscript that you could pay more attention to parallel construction in sentences, but I would try to maybe break up the sentence or otherwise clarify.

Methods:

6. This may be a my institution thing, but I don't see a reference to ethical review; in my experience all surveys require ethical review and if that is not the policy at your institution when they are not patient-facing, at least an exemption from ethical review ought to be sought and documented (editors can correct me if this is not policy Internationally)
7. My main question about the methods is about the weighting of the different governance principle; was it on purpose that strategic vision was worth 12 points, and intelligence/information only worth 2 or was that just a factor of the number of questions in each section (in which case there should have been some controlling for that before getting to the total score). . . Does that make sense? I see how your scoring came about from the specific questions in each section, but was this by design (in which case I would ask that you describe that when you talk about scoring or the tool) or should the components of governance be weighted more equally (in which case you can still start with the same number of points, but maybe you should be adding weighted scores to get the total result).
8. At the end of the first paragraph about scoring, I believe you mean "MLO" not "ML"; else does not seem clear
9. The results say there were 5 respondents from Rwanda but here it says 4; please correct

	Discussion 10. would omit "adapted" from the first line 11. third sentence would rewrite for parallel construction 12. I think you are missing "its" on page 18, line 46 about Rwanda "hosting [its] first national symposium. . ." 13. While I agree with your sentiments, when you talk about improving injury prevention in South Africa (paragraph 3), I am not sure if it is supported by your results (the only question that seems to be about focus on injury prevention is the one about seat belt laws for which South Africa got max scores); I would argue that while what you are saying is correct, perhaps it is a limitation of your survey that you did not capture more of that gap and I would maybe include this in the discussion of limitations of your survey 14. just a small point but I would consider your word choice at the end of paragraph 7 (page 21, line 32) where you say lessons "can be used for using" 15. when you say "outmost" I believe you mean "utmost"
--	--

VERSION 1 – AUTHOR RESPONSE

Reviewer Reports:

Reviewer: 1

Dr. Tamara J. Worlton, Uniformed Services University of the Health Sciences

Comments to the Author:

Excellent paper and one that can be followed up in time to see if interventions/improvement has been accomplished.

Reply: Thank you very much for your encouraging comment.

Some minor comments/edits:

In abstract introduction- the second sentence does not seem to fit with the paper. You are evaluating this so perhaps this should read as little is known or published about healthcare governance. This study aims to evaluate governance in three sub-Saharan countries from policy maker and injury care provider perspective...

Reply: Thank you for your insightful comment. This has been edited accordingly. Please see page 2 line 41-42.

“Objectives: This study aims to evaluate health systems governance for injury care in three sub-Saharan countries from policymakers' and injury care providers' perspectives.”

Abstract conclusion: All three sentences are kind of saying the same thing. I recommend linking it back to your intro- This study provides baseline insight and evaluation of governanceAreas of improvement include

Reply: We have adjusted accordingly. Please see page 3 line 68-72.

“In this multi-country governance survey, we provide insight into and evaluation of health system governance for trauma in three low and middle-income countries in Sub-Saharan Africa. It highlights

areas of improvement that need to be prioritised, such as transparency, to meet the high burden of trauma and injuries in low and middle-income countries.”

Methods: The descriptions of the countries/study setting does not belong in methods- it should be in the introduction or discussion. If you have space for another table- I would put the country characteristics in a table form (study setting) that way the reader can see more clearly the differences between the three.

I think table 1 is not needed and appendix table 1 should be used instead. When I looked at the table 1 it was not clear why each principle had a different max value.

Reply: Please see editorial comments. We would like to keep the description of the setting in the methods as we have already done as this is common in BMJ open.

Is there a reason why only the investigator score is shown in Table 3 and both scores are only shown in appendix table 3? The level of concordance between investigator and respondent scores is a big point that should be highlighted more. I believe a bar graph could highlight this better.

Reply: If we were to include the respondent scores with all the questions in table 3, the table would be too large to fit in the main document, as per journal guidelines. Therefore, we included this “overall” table to the appendices. However, we have now added appendix table 3 as table 4 in the main document, which highlights the differences between the investigator scores and the respondent scores. Please see page 17.

page 21 line 11- is this supposed to be Siddiqi vice Saddiqi?

Reply: Thanks for noticing this error. It has been adjusted accordingly. Please see page line.

Reviewer: 2

Dr. Nina Glass, Rutgers New Jersey Medical School

Comments to the Author:

Thank you for inviting me to review this manuscript. It is an important account that begins to address the topic of governance in the provision of trauma care. Overall I think that the manuscript is considered and well-written. I did have a few questions and a few editorial comments for the authors that I will outline here.

Reply: Thank you very much for your encouragement and for taking the time to review our manuscript.

Abstract:

1. It seems as if maybe you were trying to cut too many words; e.g., "as" was omitted twice from the results section (should be "such as").
2. I will come to this again in the introduction, but you have not referred to the "increasing" burden of trauma in the abstract, so not sure if you can say that in the conclusion of the abstract ("high" or "overwhelming" or something else would be fine).
3. Just editorial, but the third bullet about how the study affects practice has a typo in the header

Reply: Thanks for reading our paper in such detail. The abstract and the bullet points have been adjusted after the editorial comments, and the increasing burden of trauma has been changed to high burden of trauma throughout the manuscript.

Introduction

4. Speaking of an increasing burden of traumatic injury always raises red flags for me; is injury burden actually increasing? Is its percentage of the cause of death and disability increasing because care for other conditions is improving faster? Aren't we driving safer cars and overall getting safer or is the burden of this condition actually increasing? I'm not sure if your overview addresses this (and maybe you don't have to and this is simply my problem).

Reply: We agree with your comment. It is more a relative increase rather than an absolute increase and we have made some edits considering this comment in the introduction. Please see track changes page 6 line 138-144. Also the increasing burden has been edited to the high burden of injuries throughout.

“Whilst deaths and disability life years (DALYs) lost from many other conditions are in decline, DALYs from injuries remain stubbornly high. Indeed, deaths from injury are predicted to become the leading cause of death by 2030.³ Despite improvements in road traffic safety in most high-income countries, many LMICs are now having an increasing number of motorised vehicles and road traffic accidents in addition to other common accidents causing injuries such as falls and burns. Still, only a few LMICs have well-defined trauma systems or trauma registries.”

5. The sentence in paragraph one beginning, "The Lancet-University of Oslo Commission. . ." is way too long and therefore something about it doesn't make sense; it may be a problem I saw several times in the manuscript that you could pay more attention to parallel construction in sentences, but I would try to maybe break up the sentence or otherwise clarify.

Reply: Thanks for making us aware of this. We have made several edits, please see tracked changes throughout the manuscript.

Methods:

6. This may be a my institution thing, but I don't see a reference to ethical review; in my experience, all surveys require ethical review and if that is not the policy at your institution when they are not patient-facing, at least an exemption from ethical review ought to be sought and documented (editors can correct me if this is not policy Internationally)

Reply: The study was approved by the ethical review board at the University of Birmingham. We have added a section on ethical approval in the text. Please see page 12 line 337-339.

“The overall study was approved by University of Birmingham Research Ethics Committee, UK (ERN_20- 00880).”

7. My main question about the methods is about the weighting of the different governance principle; was it on purpose that strategic vision was worth 12 points, and intelligence/information only worth 2 or was that just a factor of the number of questions in each section (in which case there should have been some controlling for that before getting to the total score). . . Does that make sense? I see how your scoring came about from the specific questions in each section, but was this by design (in which case I would ask that you describe that when you talk about scoring or the tool) or should the components of governance be weighted more equally (in which case you can still start with the same number of points, but maybe you should be adding weighted scores to get the total result).

Reply: Each governance principle was not weighted. Each principle already had a set number of questions according to the original governance survey by Siddiqi, and this was only adapted by us to assess governance for injury and trauma care. Hence, for Strategic vision there was already eight questions developed to give the maximum score of 12, whilst for Intelligence/information there was

only one question giving a maximum score of 2. Whilst we recognise that some principles are more important hence weighting might be appropriate, at present times there is little known between presence of governance and outcomes. Hence exactly how to weight the principles appropriately is uncertain therefore we followed what was recommended by the original tool. However, to be able to compare the principles to each other, we then calculated percentage scores, so the score achieved in the survey was divided by the maximum possible score for each principle and multiplied 100%. We did this both for the respondent scores and for the weighted investigator score. We have tried to explain this better in the manuscript now. Please see page 9-10 line 270-300.

“Scoring was done separately for each country. For each principle there were already a set number of questions outlined by Siddiqi et al. to give a maximum score. Responses were awarded points for each question and treated as binary categorical (0 or 1) or ordinal (0, 1 or 2) (see appendices). We derived scores using two methods - investigator-weighted scores and respondent scores. For the respondent scores, the mean score across respondents for each question was computed as the average score from the responses for each question. Given that the response rate for each country differed, the denominator (n) varied based on the number of responses: 11 for Ghana, 5 for Rwanda and 5 for South Africa.

Whilst the investigator-weighted scores considered the following to derive a final score for each question: results from respondents, respondents’ professional roles, and the availability of evidence from policy documents and the grey literature searches. These investigator scores were derived after discussions between the authors. Consideration of the respondent’s professional roles depended on the question asked; more emphasis was given to responses from policymakers rather than trauma care providers for policy-related questions, and more was given to trauma care providers for questions related to service provision. So, for example, if a trauma care provider gave a score of 0, and the policymaker gave a score of 2 on a question related to policy, such as; “*are there legal documents of injury care?*”, the question would receive a final score of 2 as the policymaker was more likely to have up-to-date knowledge. If a policy document was available to answer a question definitively, the literature took precedence over respondents. This process was done through discussions between two authors: AMAL and MLO. When there were disagreements, third and fourth investigators served as arbiters (AI and JD).

Both investigator scores and the average respondent scores for each principle were calculated by dividing the achieved score in each country by the total score possible to achieve and multiplying it by 100. Comparisons across countries are described for each of the 10 Principles and overall.”

8. At the end of the first paragraph about scoring, I believe you mean "MLO" not "ML"; else does not seem clear

Reply: Thanks for noticing this error. It has been adjusted accordingly. Please see page 10 line 295.

9. The results say there were 5 respondents from Rwanda but here it says 4; please correct

Reply: We have corrected this to 5 respondents throughout the manuscript.

Discussion

10. would omit "adapted" from the first line

Reply: We have removed the work “adapted” from the first line of the discussion. Please see page 17 line 407-408.

11. third sentence would rewrite for parallel construction

Reply: We have rewritten the third sentence of the discussion. Please see page 17 line 411-413.

“However, considering the massive burden of injuries and trauma in these countries, our results suggest that there is room for improvement even in the higher-performing countries.”

12. I think you are missing "its" on page 18, line 46 about Rwanda "hosting [its] first national symposium. . ."

Reply: This error has been corrected. Please see page 17 line 420-421.

“This includes developing policies, training healthcare providers, investing in data collection, and hosting its first national symposium on trauma and injuries in 2019.”

13. While I agree with your sentiments, when you talk about improving injury prevention in South Africa (paragraph 3), I am not sure if it is supported by your results (the only question that seems to be about focus on injury prevention is the one about seat belt laws for which South Africa got max scores); I would argue that while what you are saying is correct, perhaps it is a limitation of your survey that you did not capture more of that gap and I would maybe include this in the discussion of limitations of your survey

Reply: We apologise that we were not clear and we have adjusted the paragraph to make it clear that we are discussing issues related to lack of governance and leadership, not injury prevention. Please see page 19 line 477-480.

“Prevention of violence and injury should be a strategic priority for government programmes and policies, and this requires governance and leadership; there are valuable lessons that South Africa can learn from its own excellent governance structures for HIV care.”

We have additionally commented on the lack of questions around injury prevention in the limitations of our discussion. Please see page 22 line 588-589.

“Another limitation was that there was only one question focusing on injury prevention in our survey.”

14. just a small point but I would consider your word choice at the end of paragraph 7 (page 21, line 32) where you say lessons "can be used for using"

Reply: We have adjusted the text accordingly. Please see page 21 line 561-563.

“ There are valuable lessons from these other disease areas that can be used for governance structures for improving trauma care systems.”

15. when you say "outmost" I believe you mean "utmost"

Reply: This word has been removed to avoid confusion.

Reviewer: 1

Competing interests of Reviewer: No competing interests to disclose

Reviewer: 2

Competing interests of Reviewer: none

VERSION 2 – REVIEW

REVIEWER	J. Worlton, Tamara Uniformed Services University of the Health Sciences
REVIEW RETURNED	17-Jul-2023

GENERAL COMMENTS	Thank you for these revisions! I have only minor editing which might have been cleared up on a non-track changes version. Table 4 is an excellent addition, thank you! line 127 missing "adjusted" in disability-adjusted life year (DALY). line 181- you already defined DALY and don't need to spell it out again. line 185, 193 - "of" is italicized. Sentence starting 197 is too long. Would end after governance (line 199) and start like " The survey consisted of 37 questions.." line 212- strike "documents" line 243- strike "whilst" line 382- should be (UHC) not (UCH) line 388 -"will" no "wil" line 415- tuberculosis is abbreviated TB, but not sure if it was used before.
--